# Direct observations of a surface eigenmode of the dayside magnetopause

M.O. Archer [1,2], H. Hietala[3,4], M.D. Hartinger[5,6], F. Plaschke [7] & V. Angelopoulos[3]

The abrupt boundary between a magnetosphere and the surrounding plasma, the magnetopause, has long been known to support surface waves. It was proposed that impulses acting on the boundary might lead to a trapping of these waves on the dayside by the ionosphere, resulting in a standing wave or eigenmode of the magnetopause surface. No direct observational evidence of this has been found to date and searches for indirect evidence have proved inconclusive, leading to speculation that this mechanism might not occur. By using fortuitous multipoint spacecraft observations during a rare isolated fast plasma jet impinging on the boundary, here we show that the resulting magnetopause motion and magnetospheric ultra-low frequency waves at well-defined frequencies are in agreement with and can only be explained by the magnetopause surface eigenmode. We therefore show through direct observations that this mechanism, which should impact upon the magnetospheric system globally, does in fact occur.

[1] School of Physics and Astronomy, Queen Mary University of London, Mile End Road, London E1 4NS, UK. [2] Space and Atmospheric Physics Group, Department of Physics, Imperial College London, South Kensington Campus, London SW7 2AZ, UK. [3] Department of Earth, Planetary and Space Sciences, University of California, Los Angeles, 595 Charles Young Drive East CA 90095-1567, USA. [4] Space Research Laboratory, Department of Physics and Astronomy, University of Turku, 20500 Turku, Finland. [5] Space Science Institute, 4750 Walnut St Suite 205, Boulder, CO 80301, USA. [6] Department of Electrical and Computer Engineering, Virginia Tech, Perry St, Blacksburg, VA 24060, USA. [7] Space Research Institute, Austrian Academy of Sciences, Schmiedlstraße 6, 8042 Graz, Austria. Correspondence and requests for materials should be addressed to M.O.A. (email: m.archer@qmul.ac.uk)

Planetary magnetic fields act as obstacles to solar/stellar winds with their interaction forming a well-defined region of space known as a magnetosphere. The outer boundary of a magnetosphere, the magnetopause, is arguably the most significant since it controls the flux of mass, energy, and momentum both into and out of the system, with the boundary's motion thus having wide ranging consequences. Magnetopause dynamics, for example, can cause loss-of-relativistic radiation belt electrons[1]; result in field-aligned currents directing energy to the ionosphere[2]; and launch numerous modes of magnetospheric ultra-low frequency (ULF) waves[3,4] that themselves transfer solar wind energy to radiation belt[5], auroral[6], and ionospheric regions[7]. On timescales greater than ~6 min Earth's magnetopause responds quasistatically to upstream changes to maintain pressure balance[8]. Simple models treating the dayside magnetopause as a driven damped harmonic oscillator arrive at similar timescales[9–11]. How the boundary reacts to changes over shorter timescales is not fully understood.

It was proposed that plasma boundaries, including the dayside magnetopause, may be able to trap impulsively excited surface wave energy forming an eigenmode of the surface itself[12]. The magnetopause surface eigenmode (MSE) therefore constitutes a standing wave pattern of the dayside magnetopause formed by the interference of surface waves propagating both parallel and anti-parallel to the magnetospheric magnetic field which reflect at the northern and southern ionospheres. Its theory has been developed using ideal incompressible magnetohydrodynamics (MHD) in a simplified box model, as depicted in Fig. 1a–c along

with expected polarisations (panels d, e)[13]. The signature of MSE within the magnetosphere should be a damped evanescent fast-mode magnetosonic wave whose perturbations could significantly penetrate the dayside magnetosphere[14]. Although this simple model neglects many factors which might preclude the possibility of MSE, global MHD simulations and applications of the theory to more representative models suggest MSE should be possible at Earth with a fundamental frequency typically less than 2 mHz[14,15]. The considerable variability of Earth's outer magnetosphere, however, might suppress MSE's excitation efficiency[16]. The simulations have largely confirmed the theorised structure and polarisations of MSE, but revealed that the relative phase of the field-aligned magnetic field perturbations differed from the box model prediction by 50°[15].

There exist numerous possible impulsive drivers of MSE including interplanetary shocks[17], solar wind pressure pulses[18], and antisunward plasma jets[19], all of which are known to result in magnetopause dynamics and magnetospheric ULF waves in general. However, no direct evidence of MSE currently exists and potential indirect evidence have largely been inconclusive. Space-based studies have evoked MSE to explain recurring frequencies of both magnetopause oscillations[20,21] and narrowband ULF waves excited by upstream jets[22], however other mechanisms could not unambiguously be ruled out and this interpretation of the results appears inconsistent with later MSE modelling[14]. Multi-instrument ground-based searches in the vicinity of the open-closed magnetic field line boundary suggest MSE do not occur[16,23]. While idealised theoretical treatments of plasmapause

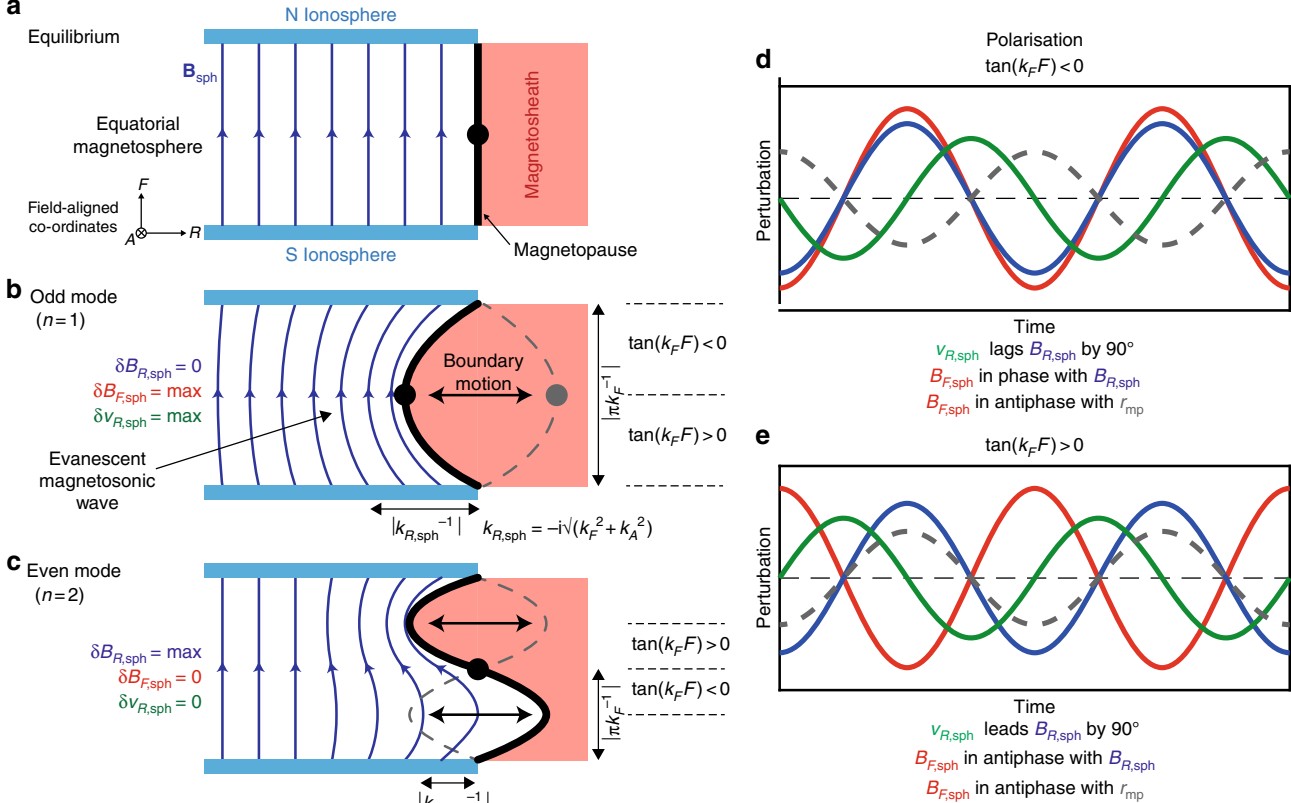

**Fig. 1** Schematic of the magnetopause surface eigenmode in a box model. **a** Box model equilibrium featuring the magnetopause (black) separating the magnetosheath (red) and magnetosphere (dark blue arrows depict the geomagnetic field bounded by the northern and southern ionospheres coloured light blue). The directions of the field-aligned coordinate system in this model are also shown where $R$ is radial, $A$ azimuthal and $F$ field-aligned. Subsequent panels depict $n = 1$ **b** and $n = 2$ **c** MSE. The midpoint of the phase is indicated as the black dot, which corresponds to the location of the MSE $n = 1$ antinode and $n = 2$ node. Expected MSE polarisations in different regions of the magnetosphere for the magnetopause standoff distance (grey dashed), radial velocity (green), radial (blue) and field-aligned (red) magnetic field components are shown on the right **d**, **e**

surface waves suggest MSE might be little affected by the ionosphere and thus observable in ground-based data[24], applications of theory specifically to MSE are currently lacking though and thus it is unclear exactly what their ground-signatures should be.

One reason perhaps why MSE, if it exists, may not have yet been observed is that impulsive drivers tend to recur on short timescales and/or are typically embedded within high levels of turbulence[17,19]. These perhaps disrupt MSE or result in complicated superpositions with various other modes of ULF wave. Evidence for other MHD eigenmodes has relied on multipoint and polarisation observations, comparing these with theory and simulations[25–27]. Therefore, multipoint observations of the magnetopause and magnetospheric response to an isolated impulsive driver may be the ideal scenario for unambiguous direct evidence of MSE.

Here we present observations at Earth's magnetosphere of an event which adhered to this strict combination of spacecraft configuration and driving conditions. We show that a rare isolated antisunward plasma jet impinged upon the magnetopause resulting in boundary oscillations and magnetospheric ULF waves. While the driving jet was impulsive and broadband, the response was narrowband at well-defined frequencies. By carefully comparing the observations with the expectations of numerous possible mechanisms, we show that the response to the jet can only be explained by the magnetopause surface eigenmode. We therefore present unambiguous direct observations of this eigenmode, which should exhibit global effects upon Earth's magnetosphere.

## Results

**Overview**. Observations are taken from the THEMIS mission on 7 August 2007 between 22:10 and 22:50 UT, a previously reported interval[28,29]. The spacecraft were ideally arranged in a string-of-pearls configuration close to the magnetopause in the mid-late morning sector and <3° northwards of the magnetic equatorial plane, as depicted in Fig. 2a, b. Subsequent panels in Fig. 2 show time-series observations in the magnetosheath (panels c, d), at the magnetopause (panels e–g), and within the magnetosphere (panels h, i). The dynamic spectra corresponding to these observations are shown in Fig. 3a–g.

**Magnetosheath observations**. THB was predominantly located in the region immediately upstream of the boundary, the magnetosheath, as evidenced by the dominance of the thermal pressure $P_{th}$ (red) over the magnetic pressure $P_B$ (blue) in Fig. 2d. At around 22:25 UT, following an outbound magnetopause crossing, THB observed an antisunward magnetosheath jet[19] lasting ~100 s with peak ion velocity ~390 km s$^{-1}$ directed approximately along the Sun-Earth line (panels a–c). An increase in the antisunward dynamic pressure $P_{dyn,x}$ and thus also the total pressure acting on the magnetopause $P_{tot,x} = P_B + P_{th} + P_{dyn,x}$ was associated with the jet (panel d). Unlike many magnetosheath jets this structure was isolated with no other significant pressure variations observed for tens of minutes afterwards[19]. The solar wind dynamic pressure was steady during this interval (grey line in panel d), with speed (average and spread) of $609 \pm 10$ km s$^{-1}$ and density of $2.7 \pm 0.1$ cm$^{-3}$. Time-frequency analysis (see Methods) revealed the jet was impulsive and broadband — power enhancements in the total pressure were contained within the jet's cone of influence with no statistically significant peaks at discrete frequencies (Fig. 3a).

**Magnetopause observations**. The magnetopause passed over four of the spacecraft (THB-E) several times. Examples of such crossings are shown in Fig. 2e, f for THC, with all crossings indicated as the coloured squares in panel g by geocentric radial distance along with the inferred magnetopause position at all times estimated through interpolation (see Methods). At least two large-amplitude ($\gtrsim 0.4$ R$_E$) inward oscillations of the boundary followed the jet. The first oscillation was largest, being observed by all four spacecraft, whereas the amplitude had already decreased by the second oscillation. The wavelet transform of the interpolated magnetopause position (Fig. 3b) shows a narrow-band enhancement in power with mean peak frequency 1.8 mHz.

Projections of the normals to the magnetopause, arrived at using the cross product technique described in the Methods section, form a fan azimuthally as shown in Fig. 2a, b. However, there was no systematic separation in direction of inbound (purple) and outbound (orange) normals. Using these normals, timing analysis was performed (described in Methods) for each inward/outward motion of the boundary. During the first inward motion of the magnetopause, concurrent with the jet, the average boundary velocity along the normal and its spread were $-238 \pm 76$ km s$^{-1}$ and showed signs of acceleration with higher velocities resulting when using later crossings. This magnetopause motion is consistent with the antisunward ion velocities of the observed magnetosheath jet (Fig. 2c). Therefore, this initial magnetopause motion was a result of the jet's impulsive enhancement in the total pressure acting on the boundary. For the subsequent magnetopause motions, the speeds were similar to one another at $24 \pm 10$ km s$^{-1}$, consistent with the 27 km s$^{-1}$ peak velocities expected for 0.4 R$_E$ sinusoidal oscillations of the boundary at 1.8 mHz. Decomposing the boundary velocities into components normal and transverse to the undisturbed magnetopause (see Methods) showed that there was little transverse motion ($8 \pm 8$ km s$^{-1}$). Indeed, the azimuthal component was consistent with zero ($-1 \pm 12$ km s$^{-1}$). No systematic differences between inbound and outbound crossings were present within these results.

At 22:22:30 UT, before the magnetosheath jet, a ~250 km s$^{-1}$ reconnection outflow[29] was observed during a magnetopause crossing (Fig. 2c), however, no further clear evidence of local reconnection occurred during subsequent crossings, likely because the observed magnetic shears were low (mean and spread were $34 \pm 22°$).

**Magnetosphere observations**. The magnetopause did not pass over THA and thus it provided uninterrupted observations of the outer magnetosphere in the vicinity of the magnetopause. The magnetic field and ion velocity observations are shown in Fig. 2h, i with corresponding wavelet spectra in Fig. 3c–g. An initial large-amplitude transient was observed immediately following the jet, chiefly in the radial components of the magnetic field $B_{R,sph}$ and ion velocity $v_{iR,sph}$ as well as the azimuthal ion velocity $v_{iA,sph}$. Longer period ULF wave activity occurred afterwards. The field-aligned magnetic field perturbation $B_{F,sph}$ showed a 1.7 mHz signal (Fig. 3e), in approximate antiphase to the magnetopause location (Fig. 2g, h). While the $B_{R,sph}$ time series appeared to exhibit a similar but opposite signal to $B_{F,sph}$ (Fig. 2h), this did not satisfy our significance test. $B_{R,sph}$ did, however, feature significant oscillations peaked at 3.3 mHz (Fig. 3c). The $v_{iR,sph}$ time series exhibited some small-amplitude complex oscillations on timescales potentially consistent with those observed in the magnetic field and boundary location (Fig. 2i), however the wavelet transform revealed no statistically significant periodicities. A clear 6.7 mHz signal dominated $v_{iA,sph}$ (Figs. 2i and 3g), a higher frequency than those previously discussed. No appreciable variations were present in $v_{iF,sph}$. Note that none of the statistically significant signals commenced before the magnetosheath jet's cone of influence (white dashed lines in Fig. 3a–g) and therefore these oscillations did not precede the jet.

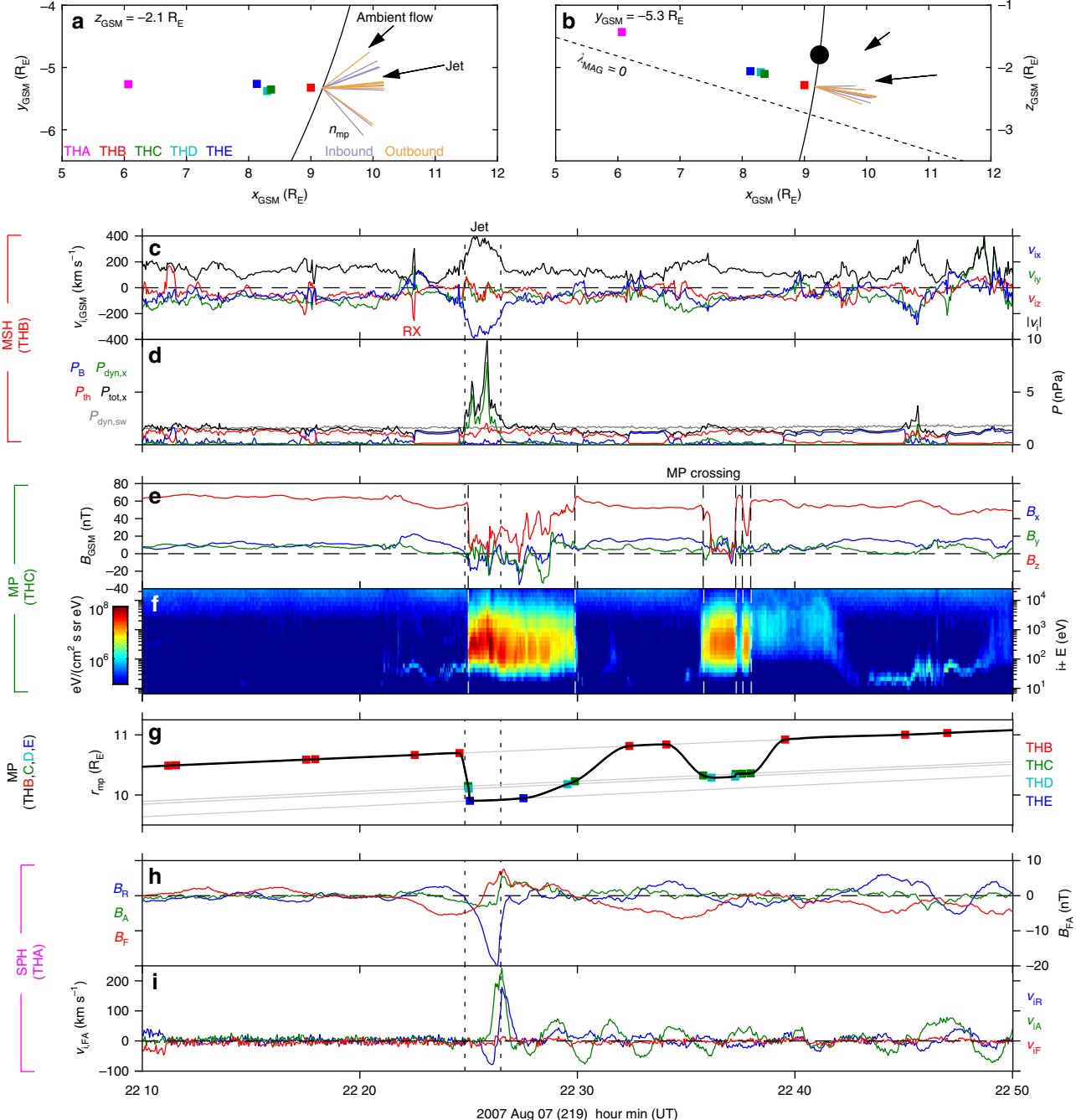

**Fig. 2** THEMIS spacecraft locations and observations. **a**, **b** Projections of the THEMIS spacecraft positions in the $z_{GSM} = -2.1\ R_E$ **a** and $y_{GSM} = -5.3\ R_E$ **b** planes. Lines indicate the model magnetopause[59] (solid) and magnetic equator (dotted). Observed magnetopause normals from inbound (purple) and outbound (orange) crossings are also shown. The black dot marks the expected location of MSE phase midpoint[14]. **c** Ion velocity at THB in GSM ($x$, $y$, $z$ as blue, green, red) and its magnitude (black). A reconnection exhaust is indicated by RX. **d** Magnetic (blue), thermal (red), antisunward dynamic (green) and total antisunward (black) pressures at THB along with lagged solar wind dynamic pressure observations by Wind (grey). **e** Magnetic field at THC in GSM (colours as before). **f** Omnidirectional ion energy flux at THC. **g** THEMIS magnetopause crossings as a function of geocentric radial distance (coloured squares) with the interpolated magnetopause location shown in black. **h** Magnetic field perturbations at THA in field-aligned (FA) co-ordinates (radial, azimuthal, field-aligned as blue, green, red). **i** Ion velocity perturbations at THA in FA co-ordinates (colours as before). Vertical dotted lines indicate times of the magnetosheath jet whereas dashed lines indicate magnetopause crossings

It is surprising that no obvious radial velocity perturbations associated with the magnetopause motion were present, regardless of whether this motion was associated with an eigenmode. However, through modelling (see Methods) we find that the expected ~27 km s$^{-1}$ amplitude velocity oscillations based on the magnetopause motion would only be detected as 6 km s$^{-1}$ due to

instrumental effects associated with cold magnetospheric ions and the spacecraft potential. The amplitude of 1.0–2.0 mHz band radial velocity perturbations were in good agreement with this, as shown in Fig. 3h.

We investigate the phase relationships between the three signals present in the THA data (Fig. 3h–k). Similar coherent

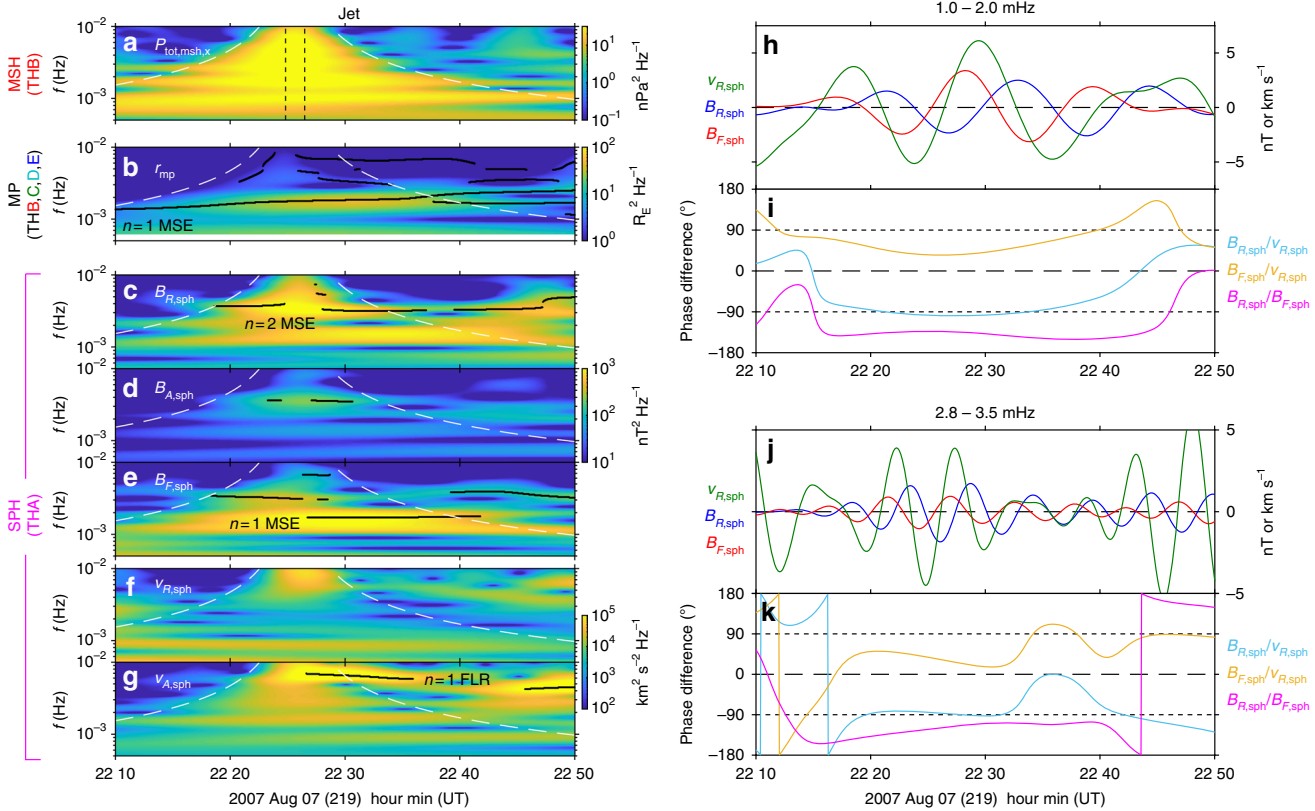

**Fig. 3** Observed dynamic spectra and phase relationships. **a–g** Wavelet dynamic power spectra of the magnetosheath total antisunward pressure **a**, magnetopause location **b**, magnetospheric radial **c**, azimuthal **d** and field-aligned **e** magnetic field perturbations, and magnetospheric radial **f** and azimuthal **g** ion velocity perturbations. Statistically significant peaks are indicated by black lines. The times of the magnetosheath jet (black dotted) and its cone of influence (white dashed) are also shown. **h–k** Wavelet band-pass filtered perturbations of the magnetospheric radial velocity (green) and radial (blue) and field-aligned (red) magnetic field pertubations at THA **h**, **j** along with their cross phases **i**, **k** where cyan is the difference between radial magnetic field and radial velocity, yellow is between the field-aligned magnetic field and radial velocity, and magenta is between the radial and field-aligned magnetic fields

phase relationships were found for the two lower frequency signals with $B_{R,sph}$ in quadrature with $v_{iR,sph}$ (means and spreads of $-96 \pm 4°$ and $-86 \pm 4°$ for the 1.0–2.0 mHz and 2.8–3.5 mHz bands, respectively) and some 50° away from antiphase with $B_{F,sph}$ ($-138 \pm 5°$ and $-123 \pm 8°$), as well as the phase between $B_{F,sph}$ and $v_{iR,sph}$ being consistent with 50° out from quadrature ($-42 \pm 8°$ and $-37 \pm 12°$). In the 4.9–8.6 mHz band $v_{iA,sph}$ led $B_{A,sph}$ by $82 \pm 6°$, likely indicating a toroidal field line resonance (FLR, a standing Alfvén wave)[27].

**Solar wind observations**. While the solar wind dynamic pressure was steady throughout this period, a number of fluctuations in the interplanetary magnetic field (IMF) were present, shown in Fig. 4b, particularly with several sign reversals in $B_{z,sw}$. Many of these fluctuations were transmitted to the magnetosheath and observed by THB, as shown in panel a where observations within the magnetosphere have been removed for clarity. It can be seen that some of these sign reversals in fact preceded the magnetosheath jet. While the magnetosheath magnetic field observations were sparse and rather turbulent, there is an apparent near one-to-one correspondence between the sign reversals in the solar wind and magnetosheath observations during the period of interest (see Methods for details of the lagging procedure). Nonetheless, we present an additional 30 min of solar wind data either side of the interval to allow for possible errors.

The magnetosheath jet occurred around the time of a magnetic field rotation which changed the IMF cone angle (the acute angle

between the IMF and the Sun-Earth line) and thus the character of the bow shock upstream of the THEMIS spacecraft. When the cone angle is below ~45° the subsolar bow shock is quasi-parallel, whereby suprathermal particles can escape far upstream leading to various nonlinear kinetic processes[30]. This results in a much more complicated shock region and turbulent magnetosheath downstream, with various transient phenomena that can impinge upon the magnetopause e.g. magnetopause surface oscillations occur more frequenctly under low cone angle conditions likely because of such transients[21]. Magnetosheath jets are just one example, with some of the strongest jets being caused by changes in the IMF orientation from quasi-perpendicular to quasi-parallel conditions[31], as appeared to be the case during this event. Following this short period of low cone angle IMF, the shock conditions were oblique or quasi-perpendicular for most of the rest of the interval.

The variations present in the upstream solar wind did not appear to be periodic. The statistical significance of the wavelet power compared to autoregressive noise is shown for the three components of the IMF (Fig. 4d–f) as well as for the solar wind density (Fig. 4h) and speed (Fig. 4j). Throughout the extended interval presented, there were very few enhancements in wavelet power for any of the quantities considered that were even locally significant (let alone the more strict global significance we have imposed on the THEMIS observations). Crucially, there were no significant enhancements peaked at (or near) either 1.7–1.8 or 3.3 mHz frequencies (indicated by the horizontal dotted lines).

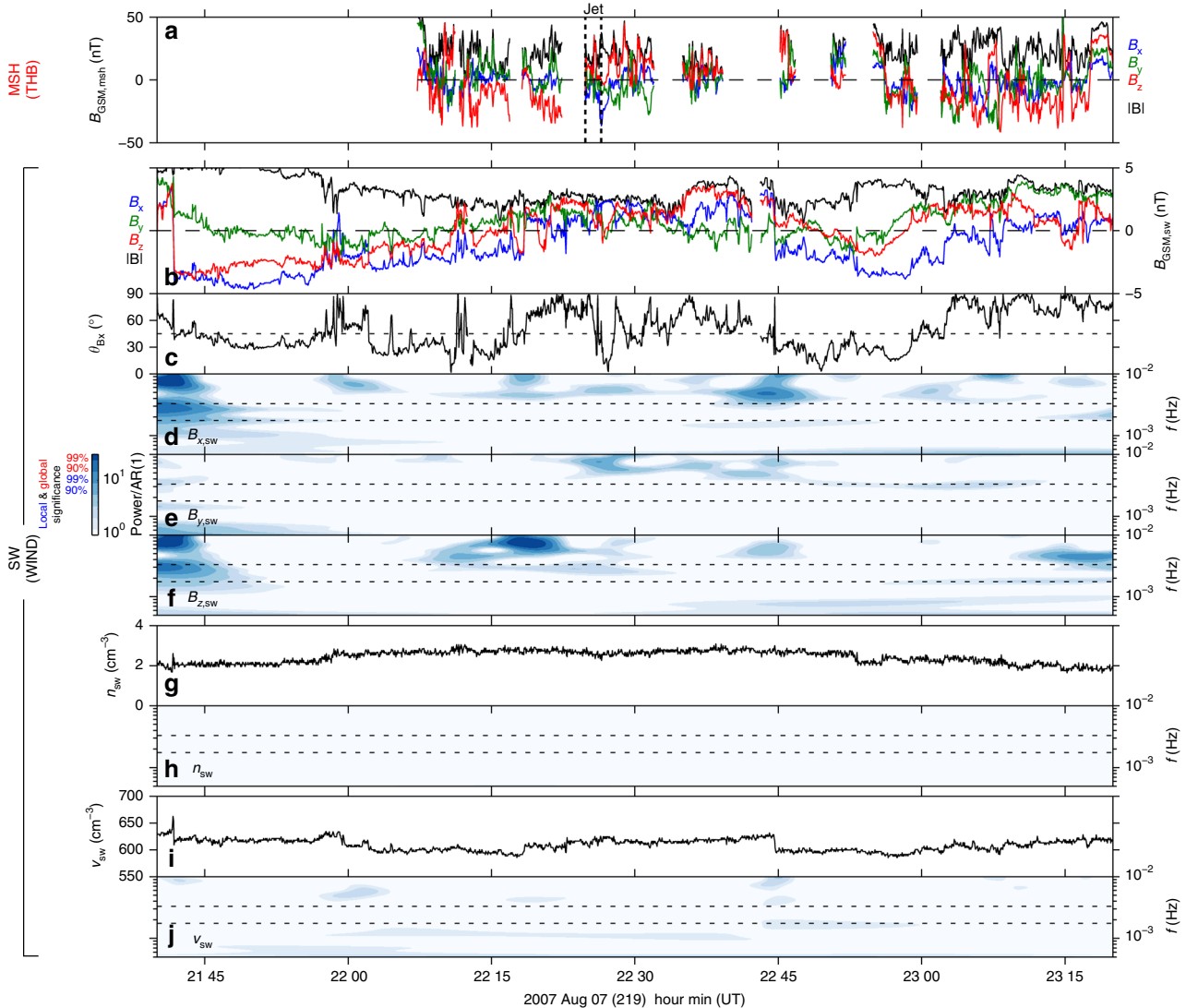

**Fig. 4** Upstream solar wind observations. **a** Magnetosheath magnetic field at THB in GSM components ($x$, $y$, $z$ as blue, green, red) and magnitude (black). Observations within the magnetosphere have been removed for clarity. The times of the magnetosheath jet are shown by vertical black dotted lines. **b–j** Lagged Wind observations of the pristine solar wind **b** magnetic field GSM components ($x$, $y$, $z$ as blue, green, red) and magnitude (black), **c** cone angle, **g** density, and **i** speed. The significance of their respective wavelet spectra are also shown **d–f**, **h**, **j**, where the power has been divided by an autoregressive noise model. Dotted horizontal lines depict frequencies of 1.7–1.8 and 3.3 mHz

Given that the aperiodic IMF variations were present before the jet but the magnetopause motions and magnetospheric ULF waves all occurred directly following it, we conclude that the magnetosheath jet was indeed the driver of the narrowband signals observed by THEMIS.

**Eigenfrequency estimates.** To aid in our interpretation of the observed signals, we compare their frequencies with estimates of various resonant ULF wave modes applied to this event using the WKB method. From an existing database of numerical calculations within representative models[14] the $n = 1$ MSE is expected at 1.4 mHz during this interval, with its antinode located at the black circle in Fig. 2b. Spacecraft potential observations from THD and THE were used to arrive at the radial profile of the electron density[32] shown in Fig. 5b (black). See Methods section for details. We combine the resulting density profile with a T96 magnetospheric magnetic field model[33,34] using hourly averaged upstream conditions, an average ion density of 6.8 amu cm$^{-3}$[35], and assuming a power law for the density distribution along the

field line using exponent 2[36]. Fundamental field line resonance (FLR) frequencies are then given at each radial distance by

$$f_{\mathrm{FLR}} = \left( 2 \int \frac{\mathrm{d}F}{v_{\mathrm{A}}} \right)^{-1}, \qquad (1)$$

where $v_{\mathrm{A}}$ is the local Alfvén speed and the integration occurs between the two footpoints of each field line, with the results shown in Fig. 5e. At THA's location this is estimated to be 6.7 mHz (panel e) in excellent agreement with the observed signal in $v_{\mathrm{iA,sph}}$, hence the observed frequency, polarisation and relative amplitudes point towards this signal being an $n = 1$ toroidal FLR.

Fast-mode resonances (FMRs), also known as cavity or waveguide modes, are radially standing fast-mode waves between boundaries and/or turning points[37,38]. In the outer magnetosphere, the lowest frequency FMRs are quarter wavelength modes resulting from over-reflection of fast-mode waves. It is thought that these may occur for magnetosheath flow speeds $\gtrsim 500$ km s$^{-1}$[39]. However, at the local times of the observations

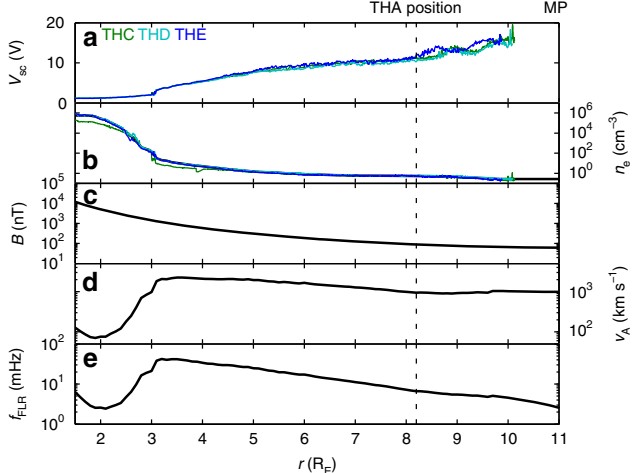

**Fig. 5** Magnetospheric radial profiles **a** spacecraft potentials, **b** potential inferred electron densities, **c** T96 magnetic field, **d** Alfvén speed, **e** fundamental field line resonance (FLR) frequency. THA's location is indicated as the dotted line

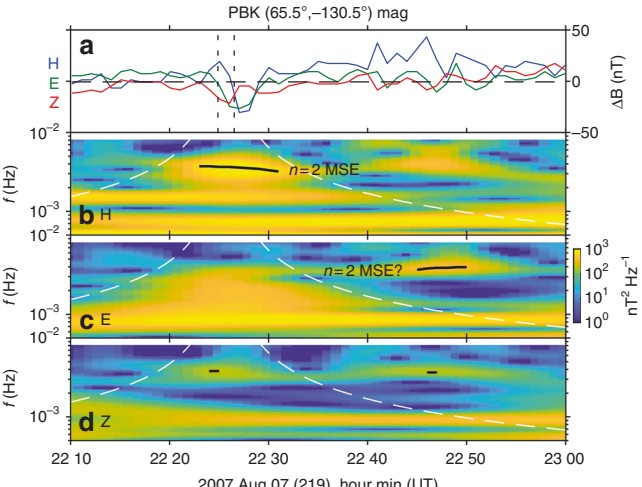

**Fig. 6** Conjugate ground magnetometer observations at Pebek **a** magnetic deflections in geomagnetic co-ordinates (H, E, Z as blue, green, red). **b**–**d** Wavelet dynamic power spectra of the H **b**, E **c** and Z **d** components in the same format as Fig. 3

this was not satisfied for either the ambient or the jet's flow speeds. Nonetheless, we still estimate the lowest possible FMR frequency given by

$$f_{FMR} = \left( 4 \int_{r_{ib}}^{r_{mp}} \frac{dR}{v_A} \right)^{-1}. \qquad (2)$$

This corresponds to a fast-mode wave propagating (assuming low plasma beta) purely in the $\pm R$ direction forming a quarter wavelength mode between the magnetopause $r_{mp}$ and an inner boundary at the Alfvén speed local maximum $r_{ib}$ (at $r = 3.2\ R_E$)[40]. From the Alfvén speed profile for this event we calculate this to be 6.3 mHz, clearly much higher than the two remaining signals which were observed.

**Ground magnetometer observations**. Unfortunately, there was very poor ground magnetometer station coverage near the spacecrafts' footpoints with only one station available, Pebek (PBK; see Methods section for selection criteria). This station was nearly conjugate with THA, whose footpoint was at (66.3°, −132.0°) geomagnetic latitude and longitude, respectively. The observations are shown in Fig. 6.

A transient, similar to that at THA immediately following the jet, was observed in the H and E components. Its timing was consistent with the ~40 s Alfvén travel time from the equatorial magnetosphere to the ground. Similar to the THA observations, following this transient other oscillations also occurred. Time-frequency analysis identified several statistically significant signals. In the H component this peaked at $3.5 \pm 0.2$ mHz and was contained within the jet's cone of influence. A later signal following the jet's cone of influence was present in the E component at $3.9 \pm 0.1$ mHz. The former was likely the ground signature of the 3.3 mHz signal observed by THA, however it is not entirely clear if this is also the case with the latter and if so why a change in polarisation occurred. Both these signals in the ground data had corresponding signatures in the Z component, though these were weak and very short lived (only 2 datapoints for each were statistically significant). While a power enhancement consistent with the 1.7–1.8 mHz signal could be seen in the H component, this did not satisfy our significance test. Finally, the 6.7 mHz toroidal FLR at THA might be expected in the

H component on the ground due to the approximate 90 rotation of Alfvén waves by the ionosphere[41]. However, its frequency was not well resolved by the coarse data being only 20% lower than the Nyquist frequency. Nonetheless, the FLR was likely the cause of the triangular wave-like oscillations present in this component following the initial transient.

The poor coverage and low resolution of the ground magnetometer data mean it is insufficient in providing additional evidence towards the physical mechanism behind the THEMIS observations.

## Discussion

We have presented THEMIS observations of the magnetopause and magnetospheric response to an isolated, impulsive antisunward magnetosheath jet. The ~100 s duration jet triggered narrowband oscillations of both the magnetopause at 1.8 mHz and magnetospheric ULF waves with peak frequencies of 1.7, 3.3, and 6.7 mHz. We now compare the observations with several possible interpretations.

(1) Direct driving. The solar wind dynamic pressure was steady throughout this interval and while there were variations present in the IMF, these were aperiodic. The magnetosheath jet's total pressure was broadband and impulsive and it has been established from the magnetopause motion and the start of the wave activity that the jet triggered the observed signals. Since no significant narrowband oscillations at (or near) these frequencies were present upstream in either the solar wind or magnetosheath, we conclude that the observed response cannot have been directly driven.

(2) Propagating Alfvén or fast-mode waves. The associated perturbations in $\mathbf{v}_{sph}$ and $\mathbf{B}_{sph}$ should either be in-phase or antiphase, unlike the observations. Furthermore, neither of these modes can explain the magnetopause motion nor the origin of the narrowband signals given the broadband driver.

(3) Propagating magnetopause surface waves. From linear analysis, the magnetospheric signature of a propagating surface wave should exhibit an in-phase/antiphase relationship between $\mathbf{v}_{sph}$ and $\mathbf{B}_{sph}$ as well as quadrature between $B_{R,sph}$ and $B_{F,sph}$[13], neither of which was observed in this event. Furthermore, while the fanning out of magnetopause

normals azimuthally is consistent with travelling surface waves, perhaps due to the Kelvin-Helmholtz instability, the lack of a difference between inbound and outbound crossings is not[42] assuming linear waves. There is no evidence from the multipoint interpolated magnetopause position for nonlinear overturning surface waves, pointing instead to a simple wave pattern. Crucially, timing analysis of the boundary (unaffected by assumptions of linearity) revealed the motions were largely directed along the normal to the undisturbed magnetopause, with azimuthal velocities consistent with zero i.e. no transverse propagation.

(4) Field line resonance. We have already concluded that the 6.7 mHz signal corresponded to a fundamental toroidal FLR at THA because of the observed polarisation and excellent agreement with the estimated frequency of this mode. The $v_{iR,\mathrm{sph}} - B_{R,\mathrm{sph}}$ phase relationships for the 1.7–1.8 and 3.3 mHz signals could be consistent with poloidal FLRs[27]. The poloidal mode is known to have slightly lower natural frequencies than the toroidal, however, these differences are typically no more than 15–30%[43]. Therefore, given that the $n = 1$ toroidal FLR frequency at THA was 6.7 mHz during this event, the much lower frequencies of 1.7–1.8 and 3.3 mHz cannot be explained as poloidal FLRs. Additionally, magnetopause motion is not expected to result from an FLR located several $R_E$ Earthward of the boundary.

(5) Fast-mode resonance. Observational signatures of radially standing fast-mode waves require ±90° phase differences between $v_{iR,\mathrm{sph}}$, equivalent to the azimuthal electric field via $\mathbf{E} = -\mathbf{v} \times \mathbf{B}$, and $B_{F,\mathrm{sph}}$[25,26], which were not observed. Exceptions to this perhaps occur in cases of exceptionally leaky or over-reflecting boundaries, however this would not be the case at the local times of the observations due to the moderate flow speeds present[39]. The large-amplitude magnetopause motions with near-zero azimuthal phase velocities are also inconsistent with a fast-mode resonance interpretation. Finally, we estimate that during this event cavity/waveguide modes of any type cannot explain frequencies below 6.3 mHz. The difference between this estimate and the observed lower frequency signals are much larger than the expected errors (~3%[44]).

(6) Pulsed reconnection. While a reconnection outflow was seen before the magnetosheath jet, no clear signatures of local magnetopause reconnection were observed subsequently throughout the event.

(7) Magnetopause surface eigenmode. The 1.4 mHz estimated fundamental MSE frequency during this period agrees with the observed 1.71.8 mHz signal within errors[14,15], with the 3.3 mHz oscillation perhaps being the second harmonic. As depicted in Fig. 1b, equatorial observations of an $n = 1$ mode should show strong signals in the motion of the magnetopause as well as $v_{iR,\mathrm{sph}}$ and $B_{F,\mathrm{sph}}$, whereas an $n = 2$ mode should dominate simply in $B_{R,\mathrm{sph}}$ (panel c). These are all in agreement with the statistically significant peaks in the wavelet spectra, after the instrumental effects on the ion velocity due to the spacecraft potential were modelled and taken into account. The similarity in observed magnetopause normals for inbound and outbound crossings as well as an azimuthal boundary velocity consistent with zero are both expected for a standing surface wave. The phase relationships between the quantities for both signals were in good agreement with theoretical expectations of MSE[13] in the regions tan $k_F F > 0$ as depicted in Fig. 1e when also taking into account the reported 50° phase shift of $B_{F,\mathrm{sph}}$ in global MHD

simulations of MSE[15]. Given the spacecraft were just southward of the expected MSE phase midpoint (Fig. 2b) this is exactly the polarisation expected for the fundamental. In contrast, the second harmonic should see the phase relations for tan $k_F F < 0$ in this region. While in the WKB approximation the $n = 1$ antinode and $n = 2$ node coincide, this may not be the case in the full solution which could exhibit anharmonicity as is the case with FLRs[36].

We therefore conclude that THEMIS observed both the $n = 1$ and $n = 2$ MSEs as the 1.7–1.8 and 3.3 mHz signals respectively, providing unambiguous direct observations of this eigenmode made possible only due to the fortuitous multispacecraft configuration during a rare isolated impulsive magnetosheath jet. MSE constitute a natural response of the dayside magnetopause, with these observations at last confirming that plasma boundaries can trap surface wave energy forming an eigenmode. Magnetopause dynamics in general have wide ranging effects throughout the entire magnetospheric system and MSE should, at the very least, act as a global source of magnetospheric ULF waves that can drive radiation belt/auroral interactions and ionospheric Joule dissipation.

It remains to be seen how often MSE occur. Future work could search the large statistical databases of magnetosheath jets for other potential events (satisfying the strict observational criteria presented in this paper) to provide further direct evidence. Other impulsive drivers could also be considered including interplanetary shocks and solar wind pressure pulses. However, since MSE are difficult to observe directly, remote sensing methods should be developed. The polarisations of magnetospheric ULF waves from spacecraft observations, as presented in this paper, may be one such method. However, potentially more useful would be ground-based signatures from magnetometers and ionospheric radar due to the wealth of data being produced. Currently, the ground signatures of MSE are not well understood, having received little theoretical attention. However, in this paper we show that MSE can exhibit at least some similar signals to the in situ spacecraft observations within conjugate high-latitude ground magnetometer data. Further investigations using theory, simulations and observations should explore all possible remote sensing methods such that the occurrence rates and properties of MSE more generally can be characterised.

## Methods

**Data.** Observations in this paper are taken from the five Time History of Events and Macroscale Interactions during Substorms (THEMIS) spacecraft[45] in particular using the Fluxgate Magnetometers (FGM)[46], Electrostatic Analysers (ESA)[47] and Electric Field Instruments (EFI)[48] all at 3 s resolution. We used the Geocentric Solar Magnetospheric (GSM) coordinate system for vector measurements from all spacecraft except THA. For this spacecraft, since we use it to evaluate the magnetospheric ULF wave response, we define a field-aligned (FA) coordinate system. The linear trend of each GSM magnetic field component was determined between 21:45 and 23:30 UT using iteratively reweighted least squares with bisquare weighting[49,50]. This trend was used to define the field-aligned direction **F** of the FA system and was subsequently subtracted from the magnetic field data. The azimuthal direction **A**, which nominally pointed eastward, was given by the cross product of **F** with the spacecraft's geocentric position. Finally the radial direction, predominantly directed radially outwards from the Earth, was determined by $\mathbf{R} = \mathbf{A} \times \mathbf{F}$. The equivalent directions of the FA system in the MSE box model are shown in Fig. 1.

Solar wind observations at the L1 Lagrange point were taken from the Wind spacecraft's 3-D Plasma and Energetic Particle Investigation[51] and Magnetic Field Investigation[52] both at 3 s resolution. In order for this data to approximately correspond to the shocked solar wind arriving in the vicinity of the magnetopause, a constant time lag was applied. First the data were time lagged by 40 min 27 s, the average amount given in the OMNI dataset from the Wind spacecraft to the bow shock nose. An additional 2 min lag to the magnetopause was subsequently added, determined by manually matching up sign reversals in the solar wind magnetic field observations with those in the magnetosheath at THB (Fig. 4a, b). Using Advanced Composition Explorer

(ACE) solar wind data instead of Wind did not substantially change any of the subsequent results.

Finally, ground magnetometer data were also used. Ground stations were chosen by computing the locations of the footpoints of the THEMIS spacecraft from a T96 model[33,34]. Only ground stations on closed field lines (according to T96) no more than 1 $R_E$ earthward from the observations and within ±1 h of magnetic local time were selected. This, unfortunately, resulted in only one station, Pebek (PBK) in the Russian Arctic. Data from this station were only available at 60 s resolution and are presented in geomagnetic co-ordinates where the horizontal components H and E point geomagnetically north and east, respectively, and Z is the vertical component. The median was subtracted from each component.

**Magnetopause motion**. To track the location and motion of the magnetopause, the innermost edge of the magnetopause current layer was identified manually from THEMIS FGM data and piecewise cubic hermite interpolating polynomials[53] were used to estimate the radial distance to the boundary from all crossings (shown as the coloured squares in Fig. 2g) at all times, resulting in the black line. This method was chosen because it does not suffer from overshooting and anomalous extrema as much as other spline interpolation methods, thus any resulting oscillations present would be underestimates. Nonetheless, the crucial aspects of the results presented, such as the time-frequency analysis, proved to be largely insensitive to the interpolation method used.

Boundary normals for each magnetopause crossing were also estimated. This was done by taking the cross product of 30 s averages of magnetic field observations either side of each crossing, which assumes that the magnetopause was a tangential discontinuity[54]. This method was used since minimum variance analysis[55] was poorly conditioned throughout the interval (the ratio of intermediate to minimum eigenvalues was ~2). The normals were insensitive to the precise averaging period used. Projections of these normals are shown in Fig. 2a, b where we distinguish between inbound and outbound crossings by colour. Magnetic shear angles were calculated from the same averaged magnetic field observations.

Finally, two-spacecraft timing analysis was also performed. Using the ascertained magnetopause normals **n**, the velocity of the boundary along the normal is given by

$$v_n = \mathbf{n} \cdot (\mathbf{r}_\alpha - \mathbf{r}_\beta)/(t_\alpha - t_\beta), \qquad (3)$$

where $r_\alpha$ is the position of spacecraft $\alpha$ during the magnetopause crossing at time $t_\alpha$. This assumes a planar surface with constant speed. For each inward/outward motion of the magnetopause, the analysis was applied to all spacecraft pairs using both sets of normals. The multiple THC crossings at around 22:37 UT were neglected. Taking the average magnetopause normal over all crossings N as representative of the undisturbed boundary, each determined magnetopause velocity can be decomposed into parallel and perpendicular velocities

$$\mathbf{v}_\parallel = v_n (\mathbf{n} \cdot \mathbf{N})\mathbf{N}, \qquad (4)$$

$$\mathbf{v}_\perp = v_n \mathbf{n} - v_n (\mathbf{n} \cdot \mathbf{N})\mathbf{N}, \qquad (5)$$

Replacing N with a normal from a model magnetopause does not significantly affect the results.

**Modelling ESA instrumental effects**. The ESA instrument can only detect ions whose energy overcomes the spacecraft potential, however the majority of ions in the magnetosphere are cold[32]. During this interval we find the temperature of cold ions to be 18 eV by fitting a Maxwell-Boltzmann distribution to the population observed in the omnidirectional ion energy spectrogram at around 22:45 UT (Fig. 2f). While no spacecraft potential observations were available for THA, those from THC-E suggest a value of ~11 V at THA's location (Fig. 5a). A sinusoidal oscillation of the magnetopause $r_{mp} = C \sin \omega t$ would result in velocity $v_{iR,sph} = C\omega \cos \omega t$ and using $C = 0.4$ $R_E$ we find that protons oscillating at 1.8 mHz would have a peak bulk kinetic energy ~4 eV, less than the assumed spacecraft potential. To estimate the effect on the data, we take one-dimensional velocity moments of the Boltzmann distribution corresponding to the cold ions, excluding all energies below the spacecraft potential. This suggests that the expected velocity oscillations of 27 kms$^{-1}$ amplitude would only be detected as 6 kms$^{-1}$ by the ESA instrument.

**Wavelet transform**. Time-frequency analysis of the data was performed using the Morlet wavelet transform[56], with the resulting dynamic power spectra shown in Fig. 3a–g. At each time all peaks between 0.5–10 mHz whose power and prominence were both above the two-tailed global 99% confidence interval (using the Bonferonni correction[57]) for an autoregressive AR(1) noise model were identified, shown as the black lines. The magnetosheath jet's cone of influence, the region within time-frequency space that is affected by the jet due to the scale-dependent windowing of the wavelet transform, are also shown as the white dashed lines. Significant narrowband signals were investigated by reconstructing a complex-numbered version of the time series from the Morlet wavelet transform across the bandwidth of each signal only[56]. The real part of the resulting time series is the band-pass filtered data whereas its phase is used to investigate polarisations. Note that it is not necessary for both time series to exhibit statistically significant power enhancements in the same region of time-frequency space for a coherent phase relationship to potentially exist between them within that region[58].

**Spacecraft potential inferred density**. The electron density can be inferred from measurements of a spacecraft's potential and in this paper we use an empirical calibration determined for THEMIS[32]. The coefficients of this calibration, however, vary from spacecraft to spacecraft and can slowly drift with time. Unfortunately, the first epoch time for these coefficients was in January 2008. Given the agreement in spacecraft potential observations with radial distance for THC-THE (the only spacecraft for which EFI was deployed shown in Fig. 5a), we simply ensure the inferred densities are consistent between spacecraft. The densities for THD and THE agreed very well, however, THC exhibited some systematic differences in density (Fig. 5b). These differences largely occurred at much smaller L-shells, nonetheless, we neglect THC density observations for this reason.

To arrive at a radial density profile, we bin the spacecraft potential inferred densities from THD and THE by radial distance using 0.1 $R_E$ bins, taking the average. The results were subsequently median filtered over 0.5 $R_E$ and the profile was extended to the model magnetopause[59] using a constant extrapolation.

## Data availability

THEMIS data and analysis software (SPEDAS) are available at http://themis.ssl.berkeley.edu. The OMNI data were obtained from the NASA/GSFC OMNIWeb interface at http://omniweb.gsfc.nasa.gov. Wind data were obtained from the NASA/GSFC CDAweb interface http://cdaweb.sci.gsfc.nasa.gov.

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

## Acknowledgements

We acknowledge valuable discussions within the International Space Science Institute (ISSI), Bern, team 350 "Jets downstream of collisionless shocks", led by F.P. and H.H. We also thank D. Burgess for helpful discussions. H.H. was supported by NASA NNX17AI45G and the Turku Collegium for Science and Medicine. M.D.H. was supported by the NASA NNX17AD35G. We acknowledge NASA contract NAS5-02099 for use of data from the THEMIS Mission. Specifically K. H. Glassmeier, U. Auster and W. Baumjohann for the use of FGM data provided under the lead of the Technical University of Braunschweig and with financial support through the German Ministry for Economy and Technology and the German Center for Aviation and Space (DLR) under contract 50 OC 0302; C. W. Carlson and J. P. McFadden for use of ESA data; D. Larson and the late R. P. Lin for use of SST data; and J. W. Bonnell and F. S. Mozer for EFI data. We acknowledge Wind plasma (courtesy of S. Bale and the late R. P. Lin) and magnetic field (courtesy of R. Lepping and A. Szabo) data. We acknowledge Oleg Troshichev and the Department of Geophysics, Arctic and Antarctic Research Institute for ground magnetometer data.

## Author contributions

M.O.A., H.H. and F.P. conceived of the study. M.O.A., H.H. and M.D.H. performed analysis on the data. M.O.A. interpreted the results and wrote the paper. V.A. gave the technical support and conceptual advice.

## Additional information

**Competing interests:** The authors declare no competing interests.

