## [Peer Review File · Nature Communications]

Reviewers' comments:

Reviewer #1 (Remarks to the Author):

Plasma waves are important for the transfer of energy and momentum into and throughout planetary magnetospheres. The study of ultra-low frequency plasma waves in Earth's magnetosphere has elucidated many of their properties and generation mechanisms. However, one category of ULF wave remains unexplained, namely apparently global waves with discrete quantized frequencies of order 1.6 mHz. One possible mechanism, a standing wave pattern on the dayside magnetopause boundary due to a Kruskal-Schwarzschild instability, was previously suggested by Plaschke et al. (2009a, b).

The current paper presents multisatellite observations which provide strong evidence for the formation of such magnetopause surface eigenmodes. The boundary waves are triggered by a localised impulsive structure ('jet') in the magnetosheath incident on the magnetopause. The alignment of the THEMIS satellites allows opportunistic detection of the boundary oscillations and associated ULF waves deeper in the magnetosphere. These are important results and I recommend publication of the manuscript. However, I have a number of comments, listed below, which should be addressed first.

1. The jet event presented in detail is one of four case studies studied by Dmitriev and Suvorova (2015). To lend confidence to their results, the authors should indicate whether similar mHz frequency magnetopause surface eigenmodes are detected for the other three case studies examined by Dmitriev and Suvorova.
2. Plaschke et al. (2009, 2013) discussed the conditions under which magnetopause surface oscillations occur most favourably, especially low cone angle. Please outline how the present observations fit with such expectations. Furthermore, Dmitriev and Suvorova noted that jets occur on average a few times per day and are an important source of magnetosheath plasma penetrating into the magnetosphere. Please mention these points in the context of the present results.
3. The discussion section of the paper outlines a number of possible mechanisms which may cause low frequency plasma waves in connection with jets. One mechanism not discussed is the over-reflection mode which may form at the near-noon magnetopause at high flow speeds, leading to quarter-wavelength mHz frequency waveguide modes (Mann et al., 1999; Walker, 2000; Wright and Mann, 2006). Mann et al. estimated that over-reflection is possible for solar wind speeds >500-600 km/s, and the waves should have similar azimuthal wavenumber (phase speed) independent of frequency. Solar wind speed for the event discussed in the present paper was around 600 km/s. Please discuss the current results in the context of the over-reflection mechanism.

4. It would be useful to have an estimate of the azimuthal phase speed for the observed wave mode, to allow comparison with expectations for Kelvin-Helmholtz and over-reflection modes.
5. It would also be useful to compare the in situ observations with ground observations, since there have been many studies with ground magnetometers and HF radars of the mHz waves which this paper associates with magnetopause surface eigenmodes. For example, an extra panel could be added to Fig. 2 or 3.
6. Minor point. Dmitriev and Suvorova (2015) represented the locations of the THEMIS satellites in terms of the nominal magnetopause position determined after Lin et al. (2010). This seems to better represent the location of THB relative to the magnetopause than in Fig. 2(a) of the current paper.

Reviewer #2 (Remarks to the Author):

Report on manuscript « First direct observations of a surface eigenmode of the dayside magnetopause » by Archer et al., submitted to Nature Communications.

This manuscript presents an event observed at the Earth magnetopause and its vicinity with the THEMIS spacecraft. This event, dated on 2007 Aug 07 by 22:25 TU is interpreted as the reaction of the magnetopause to an isolated impulsive antisunward magnetosheath jet.

The first part of the paper is devoted to data analysis. A data plot of the event is presented and commented. The magnetopause location is interpolated from the magnetopause crossings at various THEMIS satellites. A time-frequency analysis is performed with a wavelet transform. Proper modes and other characteristic frequencies associated to the global structure of the dayside magnetosphere are derived and compared to the data.

The authors show that this event, occurring during rather undisturbed conditions, is a kind of impulsional reaction of the magnetopause. Therefore, if the magnetopause has surface eigenmodes, they must appear clearly, like the sound of a drum hit once will reveal the eigenfrequencies of the instrument.

According to the authors, this is the first event showing this configuration. Usually, the magnetopause is much more constrained by various and continuous solar wind fluctuations, leading to a more intricated reaction.

This is what makes this manuscript interesting.

In the discussion, the observations are compared with seven possible interpretations.

This paper has been well prepared by previous publications by the same authors, where other types of perturbations have been studied, and theoretical derivations have been developed. The authors have a robust experience of the magnetopause dynamics.

This paper presents an interesting event, with the possibility, for the first time, of a direct measure of the eigenmodes of the magnetopause.

This paper would deserve publication, if the following points are clarified.

I suggest that the authors revise their manuscript to address specific concerns before a final decision is reached

Solar wind. Little data concerning the solar wind is presented. If we look at the ACE data (not mentioned in the MS, see figure below), we can see that the wind velocity was about 600 km/s. Therefore, at a distance of 220 RE (at the Lagrange point where ACE is located), the solar wind reaching the magnetopause at 22:25 (time of the jet) was seen $220 \times 6400 / (600 \times 60) = 39$ minutes earlier at ACE position. This corresponds to 21:46. At this time, ACE data shows an increased level of magnetic field fluctuations that lasts for about 30 minutes, including sign reversals of B_z . Because the magnetic field on THB (the spacecraft that is the first to see the jet) is not displayed, we cannot say if this 30 minutes long level of magnetic fluctuations is another cause of the observed event. In that case, the trigger would not last only 100 s, contrarily to the observed jet. Can the authors comment this point ?

Created by AMDA(c) v3.6.0 15/10/2018 16:04:25

Data

It is said on line 151 that THC observations were neglected due to systematic differences in the inferred densities.

On line 207, again a series of measures was discarded because of « instrumental effects ». Can the authors give more detailed informations about these two problems, possibly in the « Methods » section ?

Discussion

In the discussion section, the authors compare the observations with several interpretations. They show that the « magnetopause surface eigenmode » must be retained. But they discard the others, maybe a bit too fast. Here are some points that could be developed with more details.

Direct driving: do the magnetic field fluctuations seen by ACE reach spacecraft THB ? In that case, direct driving must be discussed more thoroughly.

Propagating Magnetopause surface waves: it is argued that in the case of propagating surface waves, the magnetopause normals should behave differently between inbound and outbound crossings. This is true for many types of wave, unless we have well developed surging waves. The situation is plot in the little cartoon below, where the magnetopause is drawn in black, the spacecraft motion relative to it in blue, and the inbound and outbound normals during crossing are in orange and purple. Then, we can see that the normal directions are fanning, but not alternate. Maybe the same hold for the polarization: it is not clear that a polarization computed in the linear waves approximation could hold in that highly nonlinear situation.

Field line resonance: the polarization relating $V_{IR,sph}$ and $B_{R,sph}$ is referring to paper [23]. Actually, I did not find any reference to a velocity in this paper (at least on the figures and displayed formulas). In ref [23], only the electromagnetic field is displayed. Could the authors find a better reference ? I have another problem with this paper : it shows statistics of Pc5 (assimilated to FLR) up to 9 Re only, with a high level occurrences at this distance. The THEMIS spacecrafts are at a distance of 10 RE, not so different, therefore, contrarily to what I understood in the manuscript, it is not excluded that FLR/Pc5 can be measured at the distances of the five THEMIS spacecraft.

Details

- line 65: typo in *magentic*.
- Legend of Figure 2 c: « THB velocity » could be replaced by « velocity at THB » (since THB has its own velocity that is not the data plotted on Fig 2).
- Line 254 : please correct the sentence, because cubic hermite interpolating polynomials *are* spline interpolators. So maybe you can replace « as spline interpolation » by « as other spline interpolation methods ».

Response to Reviewers

'First direct observations of a surface eigenmode of the dayside magnetopause' by Archer et al. submitted to Nature Communications

We thank both reviewers for their time reading and assessing the manuscript. We have carefully considered their reviews and have the following responses. All line numbers in the revised manuscript refer to the track changes version.

Reviewer 1

Plasma waves are important for the transfer of energy and momentum into and throughout planetary magnetospheres. The study of ultra-low frequency plasma waves in Earth's magnetosphere has elucidated many of their properties and generation mechanisms. However, one category of ULF wave remains unexplained, namely apparently global waves with discrete quantized frequencies of order 1.6 mHz. One possible mechanism, a standing wave pattern on the dayside magnetopause boundary due to a Kruskal-Schwarzschild instability, was previously suggested by Plaschke et al. (2009a, b).

The current paper presents multisatellite observations which provide strong evidence for the formation of such magnetopause surface eigenmodes. The boundary waves are triggered by a localised impulsive structure ('jet') in the magnetosheath incident on the magnetopause. The alignment of the THEMIS satellites allows opportunistic detection of the boundary oscillations and associated ULF waves deeper in the magnetosphere. These are important results and I recommend publication of the manuscript. However, I have a number of comments, listed below, which should be addressed first.

We thank the reviewer for their assessment and interest in the work.

1. The jet event presented in detail is one of four case studies studied by Dmitriev and Suvorova (2015). To lend confidence to their results, the authors should indicate whether similar mHz frequency magnetopause surface eigenmodes are detected for the other three case studies examined by Dmitriev and Suvorova.

We have looked at the other three case studies examined by Dmitriev and Suvorova (2015). We find that these events do not allow us to perform the same detailed analysis presented in the manuscript to provide direct evidence of the magnetopause surface eigenmode (MSE) – namely simultaneous observations at similar local times of:

- the driving magnetosheath jet
- the magnetopause motion by at least 2 spacecraft
- uninterrupted magnetospheric observations of ULF waves and their polarisation

We detail why this is the case for each of the three other events.

6 September 2008 16:54 UT. For this event one spacecraft (THE) observed a magnetosheath jet and another (THA) was located in the magnetosphere, however, no spacecraft observed the magnetopause following the jet, perhaps due to their respective locations. It is therefore not possible to infer the boundary's motion for this event, required for direct evidence of MSE.

5 September 2007 04:23. The spacecraft configuration was such that only one inward excursion of the magnetopause was observed by more than one spacecraft. This would be insufficient for time-frequency analysis of its interpolated position to deduce any potential discrete frequencies. Furthermore, no spacecraft remained in the magnetosphere throughout this event, restricting the ability to assess ULF waves and their phases/polarisations with the methods used in the manuscript without potentially being contaminated by edge effects associated with the magnetopause crossings.

21 July 2007 10:51. Again here no spacecraft remained in the magnetosphere throughout this event.

These additional events, therefore, would not be suitable for us to perform the same detailed analysis to provide further direct evidence of magnetopause surface eigenmodes excited by magnetosheath jets. Including these events in the manuscript would, therefore, only show that magnetosheath jets result in magnetopause motion and magnetospheric ULF waves at mHz frequencies, both of which are known results that have already been highlighted in the introduction.

It is possible that some other events listed by Dmitriev and Suvorova (2015), or the other statistical studies of Archer and Horbury (2013) and Plaschke et al. (2013), may adhere to our strict criteria. However, we feel that this would be beyond the scope of the current manuscript and could form the basis of future work. We have added a mention of this on lines 359-361.

Archer, M. O. and Horbury, T. S.: Magnetosheath dynamic pressure enhancements: occurrence and typical properties, *Ann. Geophys.*, 31, 319-331, <https://doi.org/10.5194/angeo-31-319-2013>, 2013.

Dmitriev, A. V., and A. V. Suvorova (2015), Large-scale jets in the magnetosheath and plasma penetration across the magnetopause: THEMIS observations. *J. Geophys. Res. Space Physics*, 120, 4423–4437. doi: 10.1002/2014JA020953.

Plaschke, F., Hietala, H., and Angelopoulos, V.: Anti-sunward high-speed jets in the subsolar magnetosheath, *Ann. Geophys.*, 31, 1877-1889, <https://doi.org/10.5194/angeo-31-1877-2013>, 2013.

2. Plaschke et al. (2009, 2013) discussed the conditions under which magnetopause surface oscillations occur most favourably, especially low cone angle. Please outline how the present observations fit with such expectations. Furthermore, Dmitriev and Suvorova noted that jets occur on average a few times per day and are an important source of magnetosheath plasma penetrating into the magnetosphere. Please mention these points in the context of the present results.

We have added a figure (Figure 4) which now shows the solar wind conditions, including the cone angle. This exhibited several periods indicating quasi-parallel conditions at the bow shock. In particular, the magnetosheath jet occurred around the time of an IMF rotation from quasi-perpendicular to quasi-parallel conditions, known to produce some of the strongest jets [Archer et al., 2012]. Plaschke et al. [2009] surmised that the increased occurrence of magnetopause surface oscillations under low IMF cone angle might be due to various transient phenomena resulting from the quasi-parallel shock, such as magnetosheath jets. We have added discussion of all of these points on lines 187-199.

As to the occurrence rates and other impacts of magnetosheath jets in general, we find this difficult to naturally fit within the narrative of the paper since the focus is on the magnetopause surface

eigenmode. The rare isolated magnetosheath jet presented in this paper is being used in the context of exploring MSE, rather than discussing jets more broadly. We make no claims on magnetosheath plasma penetration during this event as it is unrelated to MSE. We also note that depending on the observational criteria used, the occurrence rate of magnetosheath jets varies considerably e.g. see Plaschke et al. [2016]. We prefer to refer readers to the recent review of Plaschke et al. [2018] which highlights the current state of knowledge of magnetosheath jet generation, occurrence and impacts, including the results the reviewer mentions. We have added a reference to this on lines 64-66.

Archer, M. O., T. S. Horbury, and J. P. Eastwood (2012), Magnetosheath pressure pulses: Generation downstream of the bow shock from solar wind discontinuities, *J. Geophys. Res.*, 117, A05228, doi: 10.1029/2011JA017468.

Plaschke, F., Glassmeier, K.-H., Sibeck, D. G., Auster, H. U., Constantinescu, O. D., Angelopoulos, V., and Magnes, W.: Magnetopause surface oscillation frequencies at different solar wind conditions, *Ann. Geophys.*, 27, 4521-4532, <https://doi.org/10.5194/angeo-27-4521-2009>, 2009.

Plaschke, F., H. Hietala, V. Angelopoulos, and R. Nakamura (2016), Geoeffective jets impacting the magnetopause are very common, *J. Geophys. Res. Space Physics*, 121, 3240–3253, doi: 10.1002/2016JA022534.

Plaschke, F., Hietala, H., Archer, M. et al. *Space Sci Rev* (2018) 214: 81. <https://doi.org/10.1007/s11214-018-0516-3>

3. The discussion section of the paper outlines a number of possible mechanisms which may cause low frequency plasma waves in connection with jets. One mechanism not discussed is the over-reflection mode which may form at the near-noon magnetopause at high flow speeds, leading to quarter-wavelength mHz frequency waveguide modes (Mann et al., 1999; Walker, 2000; Wright and Mann, 2006). Mann et al. estimated that over-reflection is possible for solar wind speeds >500-600 km/s, and the waves should have similar azimuthal wavenumber (phase speed) independent of frequency. Solar wind speed for the event discussed in the present paper was around 600 km/s. Please discuss the current results in the context of the over-reflection mechanism.

This mechanism was discussed under the name of fast mode resonances. This is another term used for cavity or waveguide modes in general, though we appreciate that we should also have included these names in the manuscript as well. Most forms of fast mode resonance should exhibit a $\pm 90^\circ$ phase difference between the radial velocity (equivalent to azimuthal electric field via $\mathbf{E} = -\mathbf{v} \times \mathbf{B}$) and field-aligned magnetic field perturbations in realistic 3D geometries [Waters et al., 2002]. This phase relationship was not observed. Exceptions to this polarisation perhaps occur in cases of exceptionally leaky or over-reflecting boundaries, not considered by Waters et al. [2002].

The reviewer does raise a good point about the possibility of the over-reflection mode during this event given the solar wind speed. THB directly measured the ambient magnetosheath velocity to be ~ 140 km/s, much less than the ~ 500 km/s magnetosheath flow speeds described by Mann et al. [1999]. Indeed, even the jet's flow was less than this. This places the observations in the "moderate flow" regime of Mann and Wright [1999], hence in line with standard reflecting boundary conditions. Therefore, one would expect the $\pm 90^\circ$ phase difference in this case. We have added mention of these points on lines 233-238 and 318-321 including the relevant references.

Nonetheless, since the quarter wavelength mode between the magnetopause and the cavity's inner boundary (the Alfvén speed local maximum) will give the lowest possible frequency of any type of fast mode resonance in the outer magnetosphere, this was exactly the mode we estimated a frequency for using the observed Alfvén speed radial profile present during this event. This estimated frequency proved too high to explain the observed low frequencies.

Finally, we find the near-zero azimuthal phase speeds (see response below) are also inconsistent with the over-reflection mechanism. Our conclusion thus remains that this mode cannot explain the observed signals.

Mann, I.R and A.N. Wright, Diagnosing the excitation mechanisms of Pc5 magnetospheric flank waveguide modes and FLRs, *Geophysical Research Letters*, 26(16), 2609-2612.

Mann, I. R., A. N. Wright, K. J. Mills, and V. M. Nakariakov (1999), Excitation of magnetospheric waveguide modes by magnetosheath flows, *J. Geophys. Res.*, 104(A1), 333–353, doi: 10.1029/1998JA900026.

Waters, C. L., K. Takahashi, D.-H. Lee, and B. J. Anderson, Detection of ultralow-frequency cavity modes using spacecraft data, *J. Geophys. Res.*, 107(A10), 1284, doi: 10.1029/2001JA000224, 2002.

4. It would be useful to have an estimate of the azimuthal phase speed for the observed wave mode, to allow comparison with expectations for Kelvin-Helmholtz and over-reflection modes.

To estimate this, we have employed two-spacecraft timing analysis to the magnetopause crossings using the estimated normals, which gives the velocity along the boundary normal as

$$v_n = \mathbf{n} \cdot (\mathbf{r}_\alpha - \mathbf{r}_\beta) / (t_\alpha - t_\beta)$$

This was done for each inward and outward magnetopause motion for all spacecraft pairs and both sets of normals. While the first inward motion of the magnetopause following the jet gave speeds similar to those of the jet itself, the subsequent magnetopause motions yielded normal velocities of 24 ± 10 km/s, consistent with our estimates from the interpolated magnetopause position. Taking the average magnetopause normal over all crossings \mathbf{N} as representative of the undisturbed boundary, we can give the magnetopause velocity in terms of components parallel and perpendicular to this direction i.e.

$$\mathbf{v}_\parallel = v_n (\mathbf{n} \cdot \mathbf{N}) \mathbf{N}$$

$$\mathbf{v} = v_n \mathbf{n} - v_n (\mathbf{n} \cdot \mathbf{N}) \mathbf{N}$$

While v_\parallel remained similar to v_n at 22 ± 12 km/s, the transverse speed was small at 8 ± 8 km/s with an azimuthal (GSM y) component consistent with zero (-1 ± 12 km/s). Using a model magnetopause normal does not significantly change these results. Therefore, the observations suggest there was no azimuthal propagation of the magnetopause disturbances. This is inconsistent with the expectations for Kelvin-Helmholtz and over-reflection modes, but would be expected for a standing surface wave of the magnetopause. These points have been added on lines 122-135, 300-303, 321-323 and 425-435.

5. It would also be useful to compare the in situ observations with ground observations, since there have been many studies with ground magnetometers and HF radars of the mHz waves which this paper associates with magnetopause surface eigenmodes. For example, an extra panel could be added to Fig. 2 or 3.

Unfortunately, there was very poor ground magnetometer station coverage near the spacecrafts' footpoints (determined from the T96 model). We looked for ground stations with data that were within ± 1 hour of magnetic local time with the footpoint of THA and whose L-shells corresponded to at most $1 R_E$ earthward from the observations. THA's northern footpoint was at $(66.3^\circ, -132.0^\circ)$ geomagnetic latitude and longitude respectively with only PBK $(65.5^\circ, -130.5^\circ)$ satisfying our criteria. The only station near the southern footpoint at $(-62.6^\circ, -128.9^\circ)$ was DRV $(-74.3, -129.1^\circ)$ but, according to the T96 model, this station corresponded to open field lines. Data was only available at 60 s resolution for this event.

We now include PBK data as Figure 6. This shows the jet's initial transient effect and some signatures of the 3.3 mHz signal observed by THA. These are discussed on lines 248-273. However, the theoretical ground signatures (be they from ground magnetometers or HF radars) of MSE are not currently well understood. Future theory and simulations are required before it is possible to properly interpret observations and hence in this manuscript we simply show that some signatures of MSE can be observed on the ground, as stated on lines 367-370.

6. Minor point. Dmitriev and Suvorova (2015) represented the locations of the THEMIS satellites in terms of the nominal magnetopause position determined after Lin et al. (2010). This seems to better represent the location of THB relative to the magnetopause than in Fig. 2(a) of the current paper.

The magnetopause model in Figure 2a and 2b is only meant to be indicative and is not at all used qualitatively or quantitatively in terms of interpreting or analysing the observations presented within the manuscript. Nonetheless, using hourly averaged upstream values from the OMNI database (it is not clear from Dmitriev and Suvorova 2015 what timescale they averaged the upstream conditions over) we find little significant differences in the cuts of the two different magnetopause models as shown below, where the Shue et al. (1998) model is in black and the Lin et al. (2010) model is in blue.

Since the two models do not differ with respect to the representation of THB's location relative to the boundary in this case, we opt to keep the simpler Shue et al. (1998) model in Figure 2.

Dmitriev, A. V., and A. V. Suvorova (2015), Large-scale jets in the magnetosheath and plasma penetration across the magnetopause: THEMIS observations. *J. Geophys. Res. Space Physics*, 120, 4423–4437. doi: 10.1002/2014JA020953.

Shue, J.-H., et al. (1998), Magnetopause location under extreme solar wind conditions, J. Geophys. Res., 103(A8), 17691–17700, doi: 10.1029/98JA01103.

Lin, R. L., X. X. Zhang, S. Q. Liu, Y. L. Wang, and J. C. Gong (2010), A three-dimensional asymmetric magnetopause model, J. Geophys. Res., 115, A04207, doi: 10.1029/2009JA014235.

Reviewer 2

This manuscript presents an event observed at the Earth magnetopause and its vicinity with the THEMIS spacecraft. This event, dated on 2007 Aug 07 by 22:25 TU is interpreted as the reaction of the magnetopause to an isolated impulsive antisunward magnetosheath jet.

The first part of the paper is devoted to data analysis. A data plot of the event is presented and commented. The magnetopause location is interpolated from the magnetopause crossings at various THEMIS satellites. A time-frequency analysis is performed with a wavelet transform. Proper modes and other characteristic frequencies associated to the global structure of the dayside magnetosphere are derived and compared to the data.

The authors show that this event, occurring during rather undisturbed conditions, is a kind of impulsional reaction of the magnetopause. Therefore, if the magnetopause has surface eigenmodes, they must appear clearly, like the sound of a drum hit once will reveal the eigenfrequencies of the instrument.

According to the authors, this is the first event showing this configuration. Usually, the magnetopause is much more constrained by various and continuous solar wind fluctuations, leading to a more intricated reaction.

This is what makes this manuscript interesting.

In the discussion, the observations are compared with seven possible interpretations.

This paper has been well prepared by previous publications by the same authors, where other types of perturbations have been studied, and theoretical derivations have been developed. The authors have a robust experience of the magnetopause dynamics.

This paper presents an interesting event, with the possibility, for the first time, of a direct measure of the eigenmodes of the magnetopause.

This paper would deserve publication, if the following points are clarified.

I suggest that the authors revise their manuscript to address specific concerns before a final decision is reached.

We thank the reviewer for their assessment and interest in the work.

Solar wind. Little data concerning the solar wind is presented. If we look at the ACE data (not mentioned in the MS, see figure below), we can see that the wind velocity was about 600 km/s. Therefore, at a distance of 220 RE (at the Lagrange point where ACE is located), the solar wind reaching the magnetopause at 22:25 (time of the jet) was seen $220 \times 6400 / (600 \times 60) = 39$ minutes earlier at ACE position. This corresponds to 21:46. At this time, ACE data shows an increased level of magnetic field fluctuations that lasts for about 30 minutes, including sign reversals of Bz.

Because the magnetic field on THB (the spacecraft that is the first to see the jet) is not displayed, we cannot say if this 30 minutes long level of magnetic fluctuations is another cause of the observed event. In that case, the trigger would not last only 100 s, contrarily to the observed jet. Can the authors comment this point?

The reviewer raises a good point. We have added a figure (Figure 4) with the THB magnetosheath magnetic field observations (removing periods within the magnetosphere for clarity) comparing these to lagged solar wind data. We chose to use Wind observations because the 3DP instrument allows for plasma moments at 3 s resolution, compared to the 1 min resolution data from ACE's SWEPMAM instrument. The results prove similar for ACE though.

There was a near one-to-one association of these sign reversals from the pristine solar wind to the sparse intervals of magnetosheath observations by THB during the interval of interest, as shown in panels a-b. We include a further 30 min of solar wind data either side of the interval to account for potential errors in the lagging procedure. Nonetheless, there were sign reversals present in the THB data which preceded the jet.

Furthermore, the wavelet analysis showed that the magnetospheric ULF waves commenced within the jet's cone of influence and thus they did not precede the jet, despite the IMF variations preceding it. Finally, the timing analysis on the initial inward motion of the magnetopause matched the magnetosheath jet velocity well. Since the total pressure acting on the magnetopause increased by a factor of 5 due to the jet, it seems implausible that IMF fluctuations would be the energy source for wave activity observed to commence at the time of this impulse. We therefore conclude that the observations presented in this paper were in fact in response to the jet and not the magnetic fluctuations in the solar wind. These points have been added on lines 176-186 and 209-211.

Data. It is said on line 151 that THC observations were neglected due to systematic differences in the inferred densities. On line 207, again a series of measures was discarded because of « instrumental effects ». Can the authors give more detailed information about these two problems, possibly in the « Methods » section?

The electron density can be inferred from measurements of a spacecraft's potential and in this paper we use an empirical calibration determined for THEMIS. The coefficients of this calibration, however, vary from spacecraft to spacecraft and can slowly drift with time. Unfortunately, the first epoch time for these coefficients was in January 2008. Given the agreement in spacecraft potential observations with radial distance for THC-THE (the only spacecraft for which EFI was deployed), we simply ensure the inferred densities are in agreement between spacecraft. The inferred densities for THD and THE agreed very well, however, THC exhibited some systematic differences in density. These differences largely occurred at much smaller L-shells, nonetheless, we neglect THC observations for this reason in constructing a radial density profile for this event. This discussion has been added on lines 466-475.

The instrumental effect that the reviewer raises was one which was previously discussed in the manuscript. This concerned why no obvious radial velocity perturbations associated with the magnetopause motion were present in the THA data. This was fully explained as being due to cold ions not being detected by the ESA instrument and modelling this effect we found good agreement with the amplitudes of band-pass filtered velocity data. Therefore, no data was discarded for these reasons, but we simply explained why the signal was not as strong as one might expect. This was discussed on lines 160-166 and 437-449 though we have added a note in the discussion on line 336-337 to remind the reader that this is what we are referring to.

Discussion. In the discussion section, the authors compare the observations with several interpretations. They show that the « magnetopause surface eigenmode » must be retained. But they discard the others, maybe a bit too fast. Here are some points that could be developed with more details.

Direct driving: do the magnetic field fluctuations seen by ACE reach spacecraft THB? In that case, direct driving must be discussed more thoroughly.

As previously discussed, we have now shown that some of the fluctuations present in the pristine solar wind did reach THB. However, the IMF variations were not periodic. This is demonstrated robustly in the added figure (Figure 4) showing the wavelet transforms of the three components of the IMF (panels d-f) as well as the solar wind density (panel h) and speed (panel j). In particular, they highlight the significance of the wavelet power compared to autoregressive noise.

The reason we have changed the format of the wavelet transform presented in Figure 4 compared to the others is that it could easily be misconstrued by readers. Below we show simply the wavelet power in the three components of the IMF observations. One might consider there to be power enhancements at discrete frequencies e.g. at the times marked a and b. However, the background noise spectrum of the solar wind is not flat, thus the logarithmic colour scale can be deceptive. The small peaks in wavelet power present are far from being statistically significant and entirely indistinguishable from the expectations for noise. By normalising the wavelet power by the noise model this becomes immediately clear.

Throughout the extended interval presented in Figure 4, it is clear that there were very few enhancements in wavelet power for any of the quantities considered that were even locally significant (let alone the stricter global significance we have imposed on the THEMIS observations). Crucially, there were no significant enhancements peaked at (or near) either 1.7-1.8 or 3.3 mHz frequencies (indicated by the horizontal dotted lines). Thus showing the significance as we do in Figure 4 better highlights the null result. Similar results are obtained when using ACE data in place of Wind.

We also recall from our previous response that while the IMF variations preceded the magnetosheath jet, the magnetopause motion and ULF waves commenced at the time of the jet and thus it is highly implausible that aperiodic IMF variations could act as the energy source at this exact time. We therefore can conclude that direct driving is insufficient at explaining the observations and have added discussion of these points on lines 200-208 and 280-287.

Propagating Magnetopause surface waves: it is argued that in the case of propagating surface waves, the magnetopause normals should behave differently between inbound and outbound crossings. This is true for many types of wave, unless we have well developed surging waves. The situation is plot in the little cartoon below, where the magnetopause is drawn in black, the spacecraft motion relative to it in blue, and the inbound and outbound normals during crossing are in orange and purple. Then, we can see that the normal directions are fanning, but not alternate. Maybe the same hold for the polarization: it is not clear that a polarization computed in the linear waves approximation should hold in that highly nonlinear situation.

The reviewer raises a good point which we did not consider in the manuscript. The surging or overturning waves the reviewer describes indeed would not necessarily lead to the separation of inbound and outbound magnetopause normals that we consider to be essential for evidence of propagating surface waves along the magnetopause. While the reviewer's diagram above shows the case for such waves passing over a single spacecraft, in this study we have multiple spacecraft at widely different radial distances. Therefore, one must also consider how such overturning waves would be observed by multiple spacecraft and we modify the reviewer's diagram accordingly.

This shows that the sequence of magnetopause crossings that would be observed should also be different in the case of overturning waves. Notably, instances of spacecraft at smaller radial distances located in the magnetosheath while simultaneously other spacecraft at larger radial distances are located within the magnetosphere should occur. However, this does not fit with the observed magnetopause crossings which, as shown by its interpolated position (Figure 2g), follows a simple wave pattern with no evidence of overturning. We have added a note of this on lines 298-300.

The reviewer also raises a good point that if non-linear propagating surface waves were present then the expected polarisations derived using linear theory may not hold. We also now note this on lines 292-295.

Crucially, however, our added timing analysis (which does not rely on an approximation of linear waves) has shown that the transverse velocities of the magnetopause disturbance were small and consistent with zero azimuthally, unlike expectations for propagating surface waves. This has been added on lines 200-303.

Field line resonance: the polarization relating $V_{iR,sph}$ and $BR_{,sph}$ is referring to paper [23]. Actually, I did not find any reference to a velocity in this paper (at least on the figures and displayed formulas). In ref [23], only the electromagnetic field is displayed. Could the authors find a better reference?

In spacecraft observations ULF waves often rely on the electric field instead of the ion velocity, due to the possible instrumental effects of the spacecraft potential as mentioned earlier. Of course, these quantities are related via $\mathbf{E} = -\mathbf{v} \times \mathbf{B}$ and thus radial velocity perturbations are equivalent to those in the azimuthal electric field. We have changed the reference to Takahashi et al. (2015) who, like us, use THEMIS ion velocity measurements. Further description of the measurement technique and rationale for using velocity measurements to analyse ULF wave polarizations and Poynting vector in an MHD approximation can be found in Cummings et al. [1978].

Cummings, W. D., S. E. DeForest, and R. L. McPherron (1978), Measurements of the Poynting vector of standing hydromagnetic waves at geosynchronous orbit, *J. Geophys. Res.*, 83(A2), 697–706, doi: 10.1029/JA083iA02p00697.

Takahashi, K., M. D. Hartinger, V. Angelopoulos, and K.-H. Glassmeier (2015), A statistical study of fundamental toroidal mode standing Alfvén waves using THEMIS ion bulk velocity data, *J. Geophys. Res. Space Physics*, 120, 6474–6495, doi: 10.1002/2015JA021207.

I have another problem with this paper: it shows statistics of Pc5 (assimilated to FLR) up to 9 Re only, with a high level occurrences at this distance. The THEMIS spacecrafts are at a distance of 10 RE, not so different, therefore, contrarily to what I understood in the manuscript, it is not excluded that FLR/Pc5 can be measured at the distances of the five THEMIS spacecraft.

Of course, it is not excluded that field line resonances can have frequencies comparable to the observed signals in general. Takahashi et al. (2015) showed the L-shell dependence of fundamental toroidal field line resonances in both the morning and afternoon sectors are often only a few mHz. Similarly, Archer et al. (2015) showed using magnetospheric density measurements near noon across half a solar cycle that fundamental field line resonant frequencies are expected to often be several mHz or even sub 1 mHz.

However, for this particular event we have estimated the expected frequency of field line resonances using the T96 model and the spacecraft potential-inferred magnetospheric density, arriving at a value of 6.7 mHz at THA. Furthermore, an observed signal of 6.7 mHz at THA also demonstrated the correct polarisation for a fundamental standing toroidal Alfvén wave. Therefore, we can confidently state that the fundamental toroidal field line resonant frequency at THA for this particular event was 6.7 mHz. Poloidal field line resonances have similar but slightly lower frequencies than the toroidal mode, however, these differences are typically no more than 15-30%. Therefore, it is not possible to explain 1.7 and 3.3 mHz oscillations as poloidal field line resonances for this particular event. We have expanded the discussion to make this clearer on lines 304-312.

Archer, M. O., M. D. Hartinger, B. M. Walsh, F. Plaschke, and V. Angelopoulos (2015), Frequency variability of standing Alfvén waves excited by fast mode resonances in the outer magnetosphere, *Geophys. Res. Lett.*, 42, 10,150–10,159, doi: 10.1002/2015GL066683.

Takahashi, K., M. D. Hartinger, V. Angelopoulos, and K.-H. Glassmeier (2015), A statistical study of fundamental toroidal mode standing Alfvén waves using THEMIS ion bulk velocity data, *J. Geophys. Res. Space Physics*, 120, 6474–6495, doi: 10.1002/2015JA021207.

Details. line 65: typo in magentic.

We thank the reviewer for pointing this out and have corrected it.

Legend of Figure 2 c: « THB velocity » could be replaced by « velocity at THB » (since THB has its own velocity that is not the data plotted on Fig 2).

We have made the reviewer's suggested change.

Line 254 : please correct the sentence, because cubic hermite interpolating polynomials are spline interpolators. So maybe you can replace « as spline interpolation » by « as other spline interpolation methods ».

We have made the reviewer's suggested change.

REVIEWERS' COMMENTS:

Reviewer #1 (Remarks to the Author):

The authors have responded in detail to comments and concerns raised in the earlier review, and have revised the manuscript accordingly. This has strengthened the work. I am satisfied with the responses and revision and the manuscript should therefore be published.

Reviewer #2 (Remarks to the Author):

The authors have satisfactorily answered the questions that I raised.

I recommend the publication of the manuscript NCOMMS-18-29330A by Archer et al.